# Cell-type differential targeting of SETDB1 prevents aberrant CTCF binding, chromatin looping, and *cis*-regulatory interactions

Phoebe Lut Fei Tam [1], Ming Fung Cheung [1,2], Lu Yan Chan[1,2] & Danny Leung [1,2] ✉

SETDB1 is an essential histone methyltransferase that deposits histone H3 lysine 9 trimethylation (H3K9me3) to transcriptionally repress genes and repetitive elements. The function of differential H3K9me3 enrichment between cell-types remains unclear. Here, we demonstrate mutual exclusivity of H3K9me3 and CTCF across mouse tissues from different developmental timepoints. We analyze SETDB1 depleted cells and discover that H3K9me3 prevents aberrant CTCF binding independently of DNA methylation and H3K9me2. Such sites are enriched with SINE B2 retrotransposons. Moreover, analysis of higher-order genome architecture reveals that large chromatin structures including topologically associated domains and subnuclear compartments, remain intact in SETDB1 depleted cells. However, chromatin loops and local 3D interactions are disrupted, leading to transcriptional changes by modifying pre-existing chromatin landscapes. Specific genes with altered expression show differential interactions with dysregulated *cis*-regulatory elements. Collectively, we find that cell-type specific targets of SETDB1 maintain cellular identities by modulating CTCF binding, which shape nuclear architecture and transcriptomic networks.

Dynamic epigenomic regulation is required for establishing and maintaining precise transcriptomic networks. Histone H3 lysine 9 trimethylation (H3K9me3) is a histone post-translational modification involved in heterochromatin formation[1–4]. It participates in repressing transcription of genes and repetitive elements and regulates cellular identities during developmental progression. SET domain, bifurcated 1 (SETDB1), a member of the Suv39 subfamily of H3K9 methyltransferases (HMTases), possesses diverse functions in a plethora of cellular processes. It is critical in maintaining stem cell pluripotency and self-renewal during early development[5,6]. Absence of SETDB1 in mice is lethal at peri-implantation stages[7]. In mouse embryonic stem cells (ESCs), SETDB1 marks and silences class I and class II endogenous retroviruses (ERVs), including IAP, MLV, and ETn/MusD elements, by interacting with the corepressor Krüppel-associated box (KRAB)-associated protein 1 (KAP-1)[8–11]. A recent report found that SETDB1 also

acts in concert with NSD proteins to demarcate poised enhancers, which become reactivated upon SETDB1 loss[12]. Interestingly, SETDB1 appears to repress different genes in distinct cell-types[13–17]. Whether this phenomenon extends beyond protein-coding sequences is unclear.

Another important and malleable aspect of the epigenome is the spatial organization of chromatin. Within the nucleus, chromatin is arranged in hierarchical structures, including subnuclear compartments (A/B), topologically associated domains (TADs), and chromatin loops[18–20]. These higher-order chromatin structures function to regulate transcriptional networks in several ways[21] and their perturbation could result in molecular and whole-organism phenotypes[22,23]. An important factor in controlling 3-dimensional (3D) chromatin structures is the CCCTC-binding factor (CTCF), which mediates loop formation together with the cohesin complexes via the loop extrusion

[1]Division of Life Science, The Hong Kong University of Science and Technology, Clear Water Bay, Hong Kong, SAR, China. [2]Center for Epigenomics Research, The Hong Kong University of Science and Technology, Clear Water Bay, Hong Kong, SAR, China. ✉e-mail: dcyleung@ust.hk

mechanism[24–27]. By facilitating the interaction of distal genomic loci, CTCF can regulate gene expression through controlling physical contacts between enhancers and their target promoters[28,29]. Histone modifications have been shown to associate with chromatin structures[30,31]. Loci marked with H3K9me3 are highly enriched in the B compartments and localized to the nuclear periphery[32,33]. Moreover, in neurons, SETDB1 maintains the integrity of a specific TAD by shielding CTCF docking[34]. However, the precise involvement of H3K9me3 and SETDB1 in global chromatin organization remains elusive.

In this study, we examine H3K9me3 profiles of different mouse cells and tissues from several developmental stages and identify the mutual exclusivity of H3K9me3 and CTCF binding across tissue-types. Integrating transcriptomic, epigenomic, and chromatin structure analyses in wildtype (WT), *Setdb1* conditional knockout (KO), and *Setdb1* catalytic mutant mouse ESC lines, we discover that SETDB1/H3K9me3 regulates the 3D genome architecture by modulating CTCF binding, which is independent of DNA methylation. The loss of SETDB1 and H3K9me3 leads to a dramatic gain of CTCF binding across the genome. These increased binding sites show associated changes in chromatin accessibility and nucleosome organization. Differential CTCF binding sites are enriched with retrotransposons including SINE B2 elements, which are not previously defined as targets of SETDB1[8,16]. While global chromatin structures such as TADs and subnuclear compartments remained largely consistent, perturbation of the local chromatin interactions is detected, which is associated with the dysregulation of genes and repetitive elements. The redistribution of CTCF leads to alterations in 3D chromatin landscapes, which impacts *cis*-regulatory interactions. Notably, the impacted loci are distinct from the previously described poised enhancers co-silenced by SETDB1 and NSDs. Collectively, our results provide insights into an alternative mechanism of SETDB1 and H3K9me3 in regulating chromatin structures and transcription.

## Results

### H3K9me3 enrichment and CTCF binding are mutually exclusive across different mouse tissue-types

Previous studies have reported that H3K9me3 has diverse roles in different tissue- and cell-types. While its involvement in transcriptional repression has been well documented, recent works have suggested this histone modification and its reader proteins Heterochromatin protein 1 (HP1) could influence chromatin structures in alternative ways[31,35]. Moreover, it remains unclear as to whether the enrichment patterns and functions of this heterochromatic mark differs between cell- or tissue-types. To obtain comprehensive profiles of H3K9me3, we analysed ENCODE ChIP-seq datasets across mouse tissues from distinct developmental stages[36]. Nine mouse tissues harvested on embryonic days E14.5, E15.5, E16.5, and postnatal day P0 were included, along with our datasets from mouse ESCs (Supplementary Table 1). We detected no significant differences in the global average signal in each dataset (Supplementary Fig. 1a). However, while a small proportion of H3K9me3 peaks showed ubiquitous enrichment, the majority of peaks were divergent across tissue-types. These tissue-specific H3K9me3 enrichment patterns were retained through developmental stages (Fig. 1a).

To understand the role of the differential H3K9me3 in distinct tissues, we investigated the presence of sequence features. Transcription factor (TF) motif enrichment analysis revealed that the H3K9me3 peaks were significantly enriched with many motifs in different tissues/cells (Supplementary Fig. 1b). Some motifs corresponded to TFs with reported relationships with H3K9me3 and its HMTases. For instance, KRAB domain-containing zinc finger (KRAB-ZNF) TFs, which show lineage-specific expression, could be responsible for differential recruitment of key co-factors for SETDB1, such as KAP-1 and YY1[10,37]. Notably, we observed the significant enrichment of several TF motifs in most of the samples, including BACH1, CTCF,

NFκB, NRF2, and YY1. Integrating the corresponding RNA-seq datasets[36,38], we discovered that only CTCF was both highly expressed and had significantly enriched motifs across all tissues/cells. It was surprising that this motif, which was generally associated with active epigenetic signatures and opened chromatin states[39], was found within H3K9me3-marked regions. Given previous studies had suggested an inverse relationship between H3K9 methylation and CTCF binding at particular loci[34,40], we decided to assess the genome-wide enrichment patterns of H3K9me3 and CTCF across all available samples. Analysing all tissues at P0 and ESCs, we found a striking mutual exclusivity of H3K9me3 enrichment and CTCF binding (Fig. 1b). Loci with high CTCF signal were devoid of H3K9me3. This relationship was consistently observed across all tissues/cells. Moreover, some cell/tissue-type specific H3K9me3 peaks showed discordant CTCF binding. The same locus was either marked by H3K9me3 or CTCF in different tissues but not by both (Supplementary Fig. 1c). Taken together, our analysis suggested that H3K9me3 globally modulates CTCF binding in different tissues/cells.

### SETDB1/H3K9me3 can prevent CTCF binding independently of DNA methylation

To study the mechanistic relationship between H3K9me3 and CTCF, we conducted subsequent functional analyses in ESCs. The H3K9me3 within the facultative heterochromatin is primarily catalysed by SETDB1. Therefore, we performed ChIP-seq for H3K9me3 and CTCF in WT and *Setdb1* conditional knockout (*Setdb1* KO) cells. Strikingly, upon *Setdb1* deletion and loss of H3K9me3, a global gain of CTCF binding was detected (Fig. 2a and Supplementary Fig. 2a, b). This phenomenon was not due to elevated transcription or protein levels of CTCF (Supplementary Fig. 2c, d). The vast majority of increased CTCF peaks were observed at loci with concomitant H3K9me3 loss (2356/2669) (Supplementary Fig. 2e). Loci with increasingly significant gains of CTCF binding had concordantly larger degrees of H3K9me3 reduction (Fig. 2b). We also conducted ChIP-seq for SMC3, a subunit of the cohesin complex, in WT and *Setdb1* KO cells, which revealed similar changes (Fig. 2c and Supplementary Fig. 2f). This suggested that SETDB1/H3K9me3 played a role in hindering CTCF and cohesin binding.

To further delineate the mechanism, we conducted integrative analysis to define the changes of other epigenomic features. We discovered that in *Setdb1* KO cells, the increased CTCF binding sites were associated with DNA hypomethylation, indicated by lower levels of both 5-methylcytosine (5mC) and 5-hydroxymethylcytosine (5hmC) (Supplementary Fig. 2f). This finding was consistent with the known inhibitory role of DNA methylation on CTCF recruitment[41,42]. We noted that the consensus motif of increased CTCF peaks contained a CpG dinucleotide at position 12, which was potentially subjected to DNA methylation (Supplementary Fig. 2g). Given that previous studies have shown interplay between H3K9me3 and DNA methylation in the context of genes and repetitive elements silencing[16], we next asked if the increased CTCF binding was a consequence of DNA hypomethylation or H3K9me3 loss. We integrated ChIP-seq datasets for H3K9me3, CTCF, and SMC3 from *Dnmt* triple knockout (*Dnmt1*-/-, *Dnmt3a*-/-, *Dnmt3b*-/-) (*Dnmt* TKO) ESCs and its corresponding WT line (J1)[40]. The genome of these mutant ESCs was devoid of DNA methylation but H3K9me3 patterns are largely unchanged[16]. Remarkably, the loci with gained CTCF binding in *Setdb1* KO cells exhibited only subtle difference of CTCF enrichment in *Dnmt* TKO, similar to randomly selected CTCF peaks (Fig. 2c, d). Moreover, we confirmed that WT levels of H3K9me3 were retained in *Dnmt* TKO cells. While we did observe loci with gained CTCF binding in the *Dnmt* TKO cells, the majority were different from those defined in *Setdb1* KO, with only 6% of increased CTCF peaks in common (Supplementary Fig. 2i, j). Taken together, our results suggested that H3K9me3 can prevent aberrant CTCF binding independently of DNA methylation and the sole loss of 5mC was

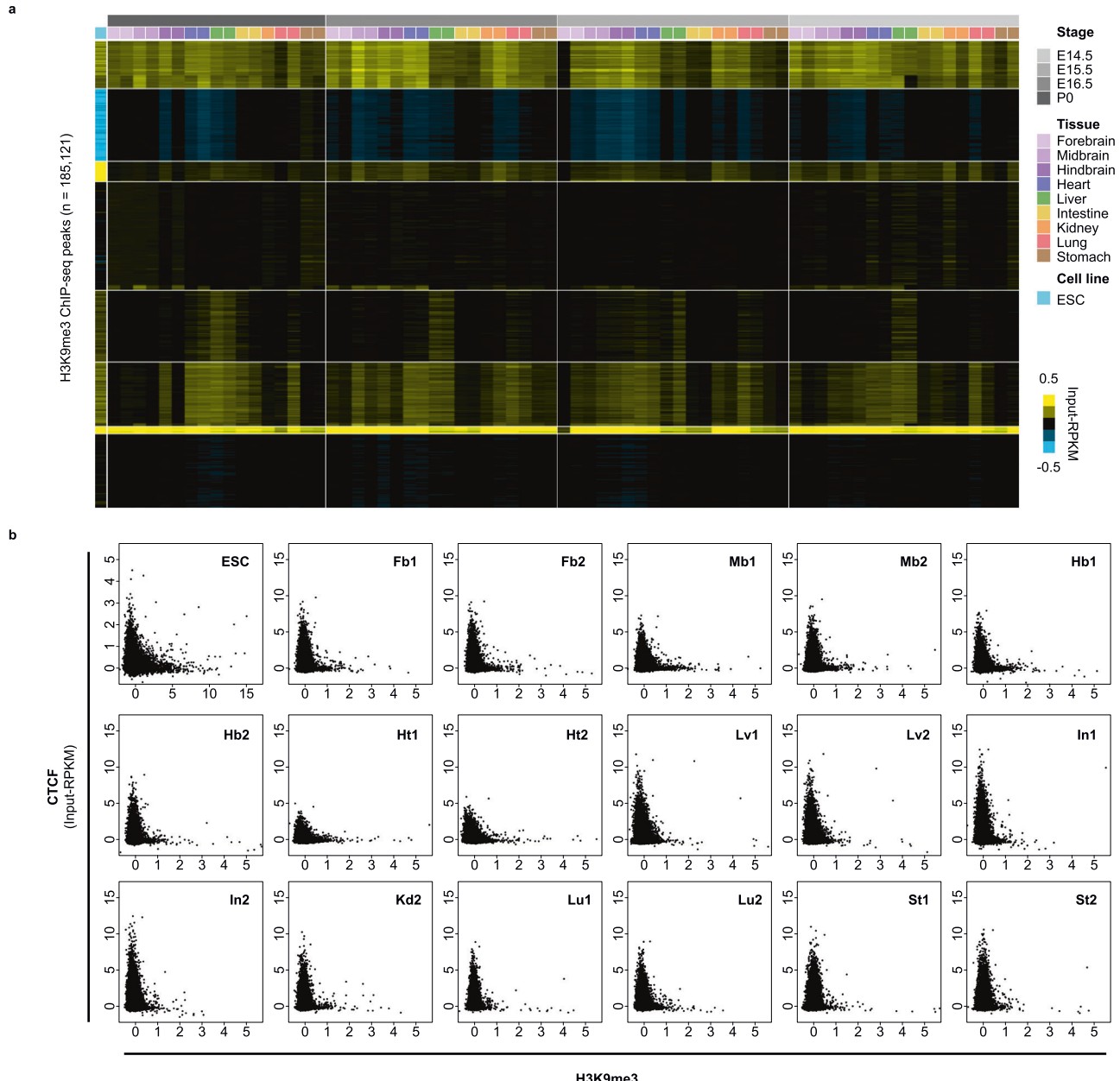

**Fig. 1 | Genome-wide analysis of H3K9me3 and CTCF enrichment in mouse ESCs, embryonic, and postnatal tissues. a** Heatmap generated by *k*-means clustering (*k* = 8) shows differential H3K9me3 enrichment in different mouse tissue- and cell-types (*n* = 10) at different developmental timepoints (embryonic day E14.5, E15.5, E16.5, and postnatal day P0). All H3K9me3 peaks defined in each sample are included and merged (*n* = 185,121). ChIP-seq signals are displayed as input subtracted Reads Per Kilobase pairs per Million reads (input-RPKM). **b** Scatter plots compare the H3K9me3 and CTCF ChIP-seq signals (input-RPKM) in ESCs and mouse P0 tissues. With the exception of ESCs and kidney, 2 biological replicates were compared for each tissue or cell-type. Each data point denotes a 5 kb window that contains both H3K9me3 and CTCF peaks defined in at least one sample (*n* = 47,172). Overall, we found a mutually exclusive relationship between H3K9me3 and CTCF enrichment. Tissue-types included are Forebrain (Fb), Midbrain (Mb), Hindbrain (Hb), Heart (Ht), Liver (Lv), Intestine (In), Kidney (Kd), Lung (Lu), and Stomach (St). Source data are provided as a Source Data file.

insufficient to induce the observed changes at H3K9me3-marked regions.

To determine if this phenomenon is dependent on SETDB1's structural presence or on H3K9me3, we performed H3K9me3 and CTCF ChIP-seq in two ESC lines harbouring catalytic mutant versions of SETDB1 (2 clonal lines with *Setdb1*$^{C1243A}$)[8]. Those peaks that showed increased CTCF binding in *Setdb1* KO cells, similarly exhibited higher ChIP-seq signals in both of the catalytic mutant lines (Fig. 2c and Supplementary Fig. 2b). As H3K9me3 could lead to chromatin compaction by recruiting HP1, we also analysed publicly available HP1 ChIP-

seq data[43]. Indeed, we found that loci with gained CTCF binding upon SETDB1 depletion, possessed higher HPγ enrichment levels in WT ESCs (Fig. 2e). Taken together, the results revealed that H3K9me3, at least in part via HP1 recruitment, functions to antagonize CTCF binding to particular sites.

Since the loss of H3K9me3 could cause alterations to chromatin states, we conducted ATAC-seq in WT and *Setdb1* KO ESCs. Concordant with its binding characteristics, CTCF peaks with increased signal upon *Setdb1* deletion also significantly gained chromatin accessibility (Supplementary Fig. 2f). We proposed that in WT cells,

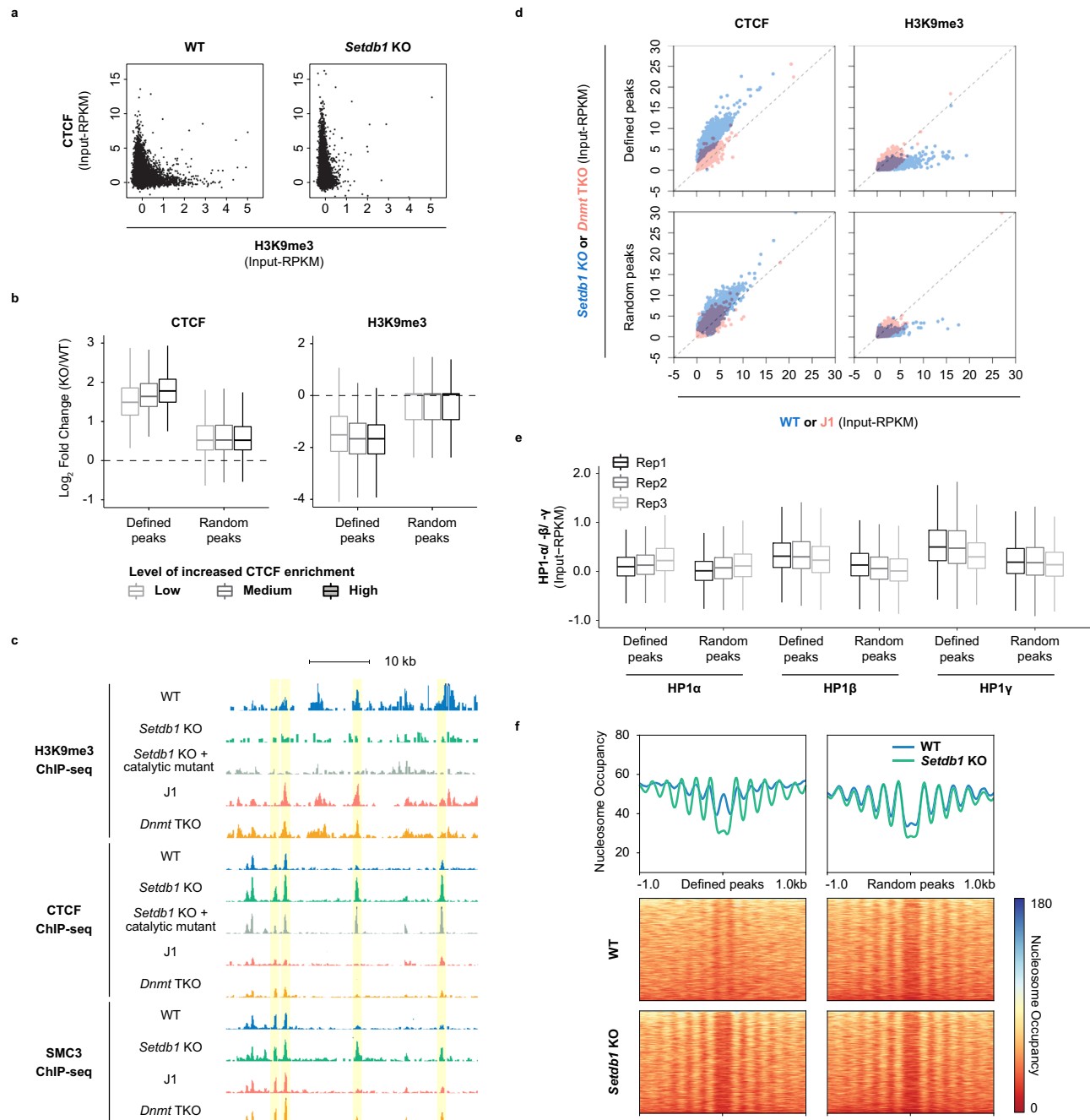

**Fig. 2 | SETDB1/H3K9me3-dependent regulation of CTCF binding. a** Scatter plots illustrate H3K9me3 and CTCF ChIP-seq signals (input-RPKM) in WT and *Setdb1* KO ESCs. Each data point indicates a 5 kb window with co-occupancy of H3K9me3 and CTCF peaks in either cell line. **b** Box plots show the association of increased CTCF binding and reduced H3K9me3 enrichment. The degree of change is represented as log$_2$-transformed fold change (*Setdb1* KO/WT) of CTCF and H3K9me3 ChIP-seq signal (RPKM) at increased (defined) or number-matched random CTCF peaks. The three incremental levels of increased CTCF enrichment are based on increasing significance calculated by one-tailed Fisher's Exact test (low: $q < 5e-4$, $n = 2669$; medium: $q < 1e-5$, $n = 1126$; high: $q < 1e-7$, $n = 445$). Boxes contain the 25th to 75th percentiles of dataset, the centre line denotes the median, and the whiskers show the lowest and highest values without outliers. **c** A genome browser screenshot demonstrates regions with H3K9me3 loss-dependent increased CTCF binding (yellow shading) on chromosome 5 (chr5:147,110,000-147,150,000), along with

increased SMC3 signal. No notable gain of CTCF binding is observed in *Dnmt* TKO. H3K9me3, CTCF, and SMC3 ChIP-seq tracks are displayed as input subtracted RPKM values, with y-axis ranging from 0–5, 0–10 and 0–15, respectively. **d** Scatter plot shows a substantial increase of CTCF signal in *Setdb1* KO but not in *Dnmt* TKO. Each dot denotes an increased (defined) or number-matched random CTCF peak ($n = 2669$). **e** Box plot illustrates increased CTCF binding sites are originally more compact with HP1γ enrichment. HP1α, β, and γ ChIP-seq signals are displayed as input subtracted RPKM in WT mouse ESCs[43] at increased and number-matched random CTCF peaks ($n = 2669$). Boxes contain the 25th to 75th percentiles of dataset, the centre line denotes the median, and the whiskers show the lowest and highest values without outliers. **f** Aggregation plots and heatmaps of MNase-seq data show the patterns of nucleosome occupancy (normalized reads) surrounding (±1 kb) increased or number matched random CTCF peaks ($n = 2669$) in WT and *Setdb1* KO ESCs. Source data are provided as a Source Data file.

H3K9me3-marked nucleosomes within the region are compacted and block CTCF binding. Therefore, we carried out MNase-seq to measure changes in nucleosome positioning. Among the increased CTCF binding sites, WT ESCs had higher nucleosome occupancy within the peak and no clear patterns of organization of the flanking nucleosomes. Upon loss of H3K9me3, these loci became depleted of nucleosomes and thus more accessible. We also observed a reorganization of nearby nucleosomes with an evident oscillating pattern, previously described to be associated with CTCF binding to chromatin[44,45] (Fig. 2f). Together, these data revealed the impact on chromatin states accompanied by dysregulated CTCF binding upon SETDB1/H3K9me3 loss.

### SETDB1 represses SINE B2 elements by blocking CTCF binding

Transposable elements are known to contribute to genomic evolution by providing a repertoire of novel TF binding sites[46–49]. Specifically, short nuclear interspersed element (SINE) B2-B4 subfamilies were shown to contribute species-specific CTCF binding motifs[20,50,51]. Given that one of the primary functions of SETDB1 in ESCs was in silencing repetitive elements, we asked whether the increased CTCF binding was occurring at derepressed retrotransposons by assessing the

underlying sequence features of CTCF ChIP-seq peaks. As expected, among all CTCF peaks, SINE elements were significantly enriched (Fig. 3a). When considering increased CTCF peaks in *Setdb1* KO cells, both SINE B2 and ERVL-MaLR elements were significantly enriched, with SINE B2 elements being drastically higher in abundance (42%) as compared to the randomly selected CTCF peaks (32%) (Fig. 3a). We then examined the epigenetic profiles of repetitive elements with increased CTCF binding upon *Setdb1* deletion (Fig. 3b and Supplementary Fig. 3a). Consistent with previous studies, we found that specific class I and II ERV subfamilies were marked by SETDB1-mediated H3K9me3 in WT ESCs (Supplementary Fig. 3b)[8,16]. Intriguingly, SINE B2 elements showed the most prominent loss of H3K9me3 in *Setdb1* KO cells (Fig. 3b). Yet, no significant changes in transcription or H3K27ac enrichment were detected (Fig. 3c). On the other hand, increased chromatin accessibility, CTCF and SMC3 binding, reduction of DNA methylation, and 5hmC enrichment were observed (Fig. 3c and Supplementary Fig. 3c). These SINE B2 elements in WT ESCs were also heavily occupied by nucleosomes (Fig. 3d). Loss of SETDB1 resulted in both nucleosome depletion within and nucleosome reorganization flanking the sequences. It is noteworthy that SINEs, including B2 elements were not previously considered among repeats repressed by

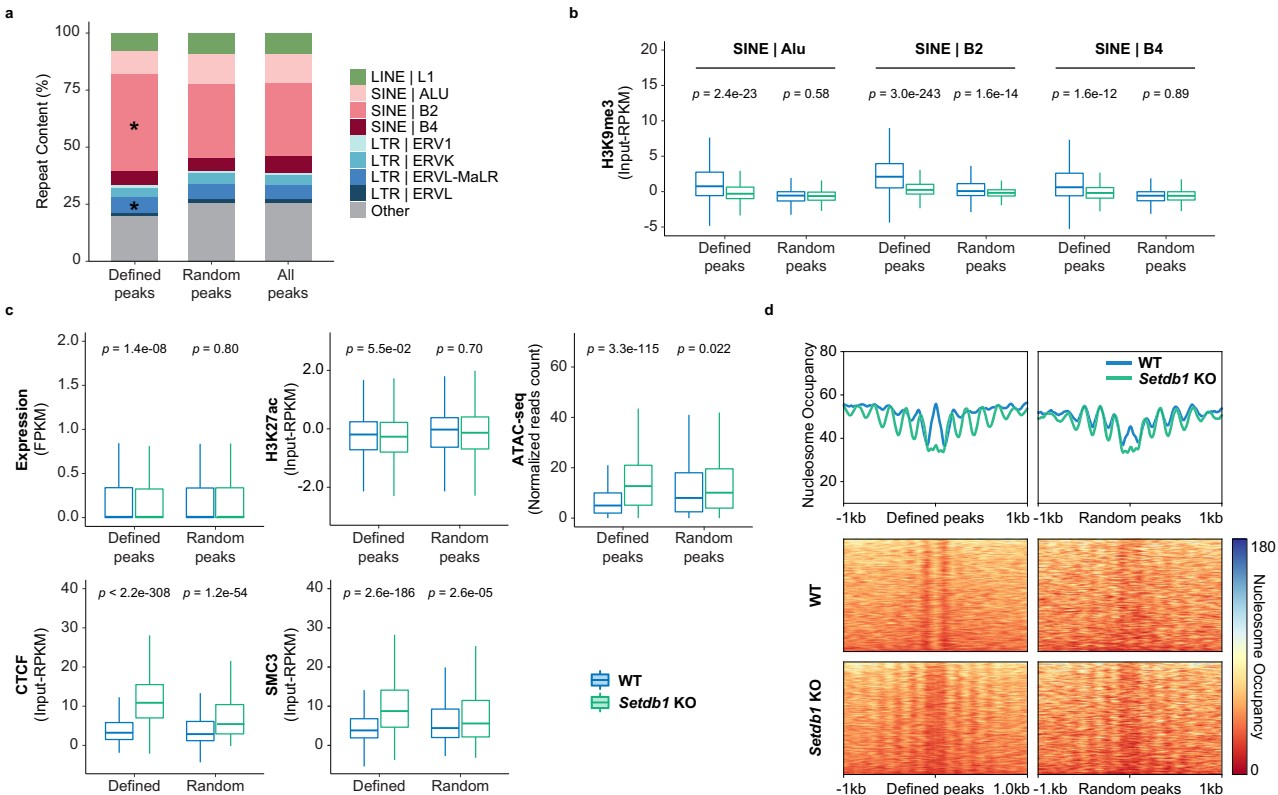

**Fig. 3 | SINE B2 elements are enriched at increased CTCF binding sites. a** Stacked bar chart shows the proportion of transposable elements located at increased and number-matched random CTCF peaks (n = 2669). Major classes of retrotransposons are represented in different colours. The overall distribution of transposable elements at all CTCF peaks (n = 61,017) is also shown as reference. SINE B2 (p = 2.13e−67) and ERVL-MaLR (p = 5.95e−5) elements are significantly enriched. Asterisk indicates p value <1e−4 calculated by two-tailed hypergeometric distribution. **b**) Box plot illustrates H3K9me3 ChIP-seq signal (input-RPKM) at Alu, B2, and B4 SINE elements located at increased (defined) and number-matched random CTCF peaks (n = 2669) in WT and *Setdb1* KO ESCs. *P* value is calculated by two-tailed Wilcoxon test. The centre and bounds of boxes refer to the median and quartiles of all data points, respectively. The minima and maxima of boxplots indicate Quartile 1 − 1.5 × interquartile range and Quartile 3 + 1.5 × interquartile range, respectively. **c**) Box plots demonstrates RNA-seq signal in Fragments Per

Kilobase pairs per Million reads (FPKM), H3K27ac, CTCF, and SMC3 ChIP-seq signals (input-RPKM), and ATAC-seq signal (normalized reads count) of SINE B2 elements located at increased and number-matched random CTCF peaks (n = 2669) in WT and *Setdb1* KO ESCs. Significance is measured by Paired Wilcoxon test. While only subtle expression and H3K27ac changes were noted, dramatic gains of chromatin accessibility, CTCF, and SMC3 binding was detected in *Setdb1* KO cells. *P* value is calculated by two-tailed Wilcoxon test. The centre and bounds of boxes refer to the median and quartiles of all data points, respectively. The minima and maxima of boxplots indicate Quartile 1 − 1.5 × interquartile range and Quartile 3 + 1.5 × interquartile range, respectively. **d** Aggregation plots and heatmaps of MNase-seq datasets show patterns of nucleosome occupancy (normalized reads) surrounding (±1 kb) SINE B2 elements that overlap with increased or number-matched random CTCF peaks (n = 2669) in WT and *Setdb1* KO ESCs. Source data are provided as a Source Data file.

SETDB1, due to a lack of transcriptional reactivation or gaining of active marks in SETDB1 depleted cells. However, our data uncovered that SETDB1/H3K9me3 functioned to repress SINE B2 elements by blocking CTCF binding. Furthermore, we observed that in WT ESCs, these SINE B2 elements are significantly enriched with KAP-1, a well-established recruiter of SETDB1 to its targets[52] (Supplementary Fig. 3d). The KAP-1/KRAB-ZNF method of targeting retrotransposons for silencing is likely how SETDB1 is directed to these specific elements.

## Subnuclear compartments and topologically associated domains are largely maintained upon global H3K9me3 loss

SETDB1 and H3K9me3 had been implicated in multiple aspects of higher-order chromatin structures. In particular, H3K9me3 correlated with the subnuclear compartmentalization definitions (A/B)[30], where B compartments are associated with nuclear periphery localization and transcriptionally inert heterochromatin[19,32,33,53]. Recent studies have shown that SETDB1 functioned with other HMTases to maintain a subset of B compartment definitions[54]. Moreover, ectopic H3K9me3 deposition at specific regions was sufficient to induce rearrangement of chromatin compartments[31]. However, whether the loss of SETDB1/H3K9me3 resulted in widescale subnuclear compartment disruption remained unclear. To address this question and to determine the state of other higher-order chromatin structures, we conducted Hi-C in WT and *Setdb1* KO ESCs. For each sample, two replicates were generated, with 2 billion raw read pairs sequenced per replicate. We obtained high quality Hi-C contacts (MAPQ ≥ 30) from over 80% of the uniquely

aligned read pairs (Supplementary Data file 1). Replicates demonstrated high degree of reproducibility and all subsequent analyses were done with replicate-merged datasets.

We segmented the genome at 25 kb resolution and demarcated A/B compartments (Fig. 4a). As expected, we found most regions with H3K9me3 enrichment resided in B compartments (Supplementary Fig. 4a). We also included a previously published H3K9me3 ChIP-seq dataset from another WT ESC line (R1)[55], which confirmed similar patterns in our B compartment definitions. The majority of nuclear compartments were unchanged in *Setdb1* KO ESCs (Fig. 4a and Supplementary Fig. 4b). Only 8.8% of the bins exhibited compartment switching. In particular, 43% of the B-to-A-switching bins were coupled with significant loss of H3K9me3 (Supplementary Fig. 4a, b). Since A and B compartments are often associated with active and inactive chromatin states, respectively, we asked if switched compartment showed consistent changes of transcription. Within bins undergoing A-to-B compartment switching, the overall expression of genes and repeats did not show substantial changes. However, the expressions of those genes and repeats in B-to-A compartment switching bins were significantly increased. This transcriptional upregulation was coupled with H3K9me3 loss (Supplementary Fig. 4c). These are likely the direct targets of SETDB1/H3K9me3-mediated silencing. Expanding to all bins that contained dysregulated genes and repeats, we found no significant changes in compartmentalization. Over 97% of transcriptionally downregulated genes and repeats exhibited no change, while 9.7% and 40.7% of upregulated genes and repeats underwent B-

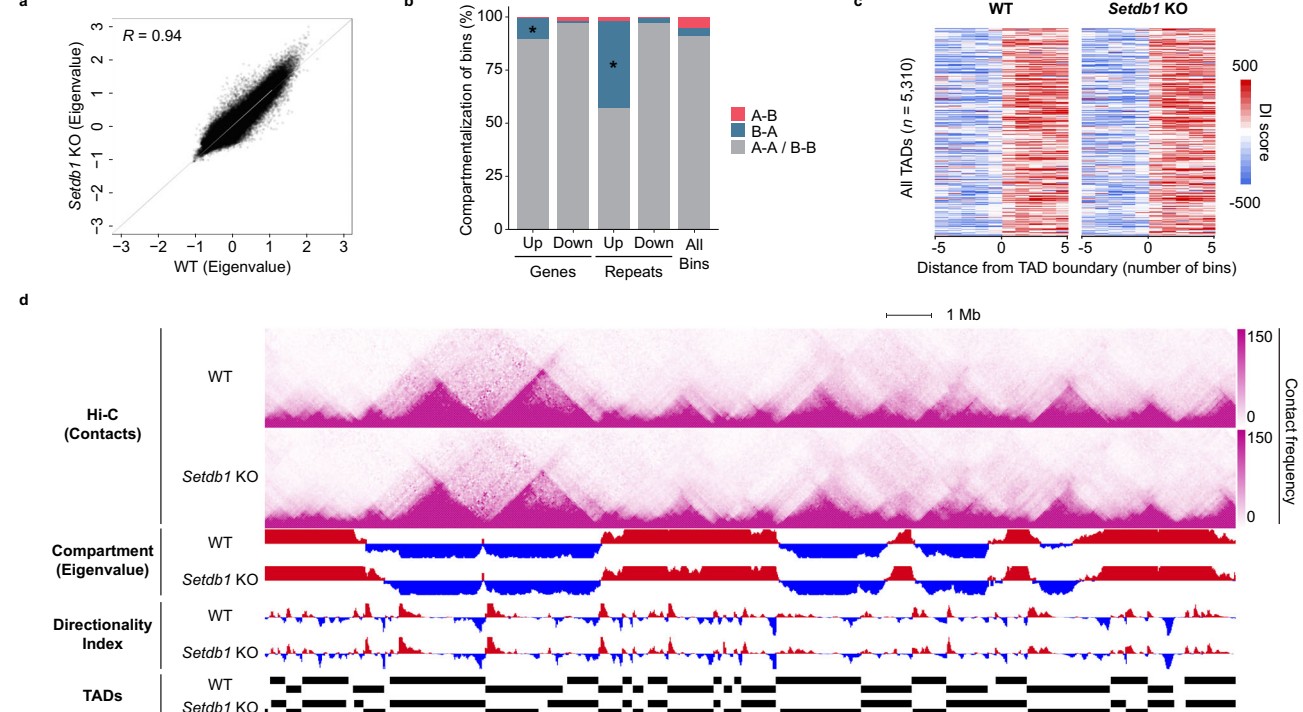

**Fig. 4 | Conserved A/B compartment and TAD definitions upon loss of SETDB1.**
**a** Scatter plot shows consistent A/B compartmentalization across the whole genome in WT and *Setdb1* KO ESCs. Each dot denotes a 25 kb window (*n* = 101,988) with its corresponding PC1 eigenvalues displayed. Pearson correlation coefficient (*R*) is included. **b** Stacked bar chart shows the proportion of compartments (25 kb bins) that undergo switching from A to B (red), from B to A (blue), and those that remain unchanged (grey). Bins are further divided by whether they contain dysregulated genes or repeats upon H3K9me3 depletion (Upregulated genes: *n* = 639; downregulated genes: *n* = 278; upregulated repeats: *n* = 6127; downregulated repeats: *n* = 1389). We detected significant enrichment of upregulated genes (*p* = 2.28e−12) and repeats (*p* < 1e-307) with concomitant B to A compartment switching in *Setdb1*

KO cells. Asterisk indicates *p* value < 1e−4 measured by two-tailed hypergeometric distribution. **c** Heatmaps show comparable TAD characteristics in WT and *Setdb1* KO ESCs. The directionality index score of ±5 bins (bin size of 25 kb) from a merged list of defined TAD boundaries (*n* = 5310) are shown. **d** A genome browser screenshot illustrates consistent compartments (y-axis of eigenvalue: −0.5−0.5) and TADs (y-axis of directionality index score: −1000−1000) across a large region on chromosome 6 (chr6:124,700,000-147,800,000) in WT and *Setdb1* KO ESCs. Hi-C contact maps also show preservation of triangular TAD structures. Hi-C contact maps display the raw counts in 50 kb windows. Positive (red) and negative (blue) eigenvalues represent compartment A and B, respectively. Black bars denote TAD definitions called by Hi-C Domain Caller. Source data are provided as a Source Data file.

to-A compartment switching, respectively (Fig. 4b). It should be noted that the overall change in eigenvalue was subtle (Supplementary Fig. 4d). This finding was intriguing, as studies have shown that modulating the transcription of particular genes was sufficient to change nuclear compartmentalization[56]. Our results confirmed that to be true at some loci, while for the majority of the genome, transcriptional upregulation was insufficient for compartment switching.

Next, to investigated if TADs were affected under global loss of H3K9me3, we applied Hi-C Domain Caller to define TADs from Hi-C datasets from each cell line (WT: $n = 2988$; *Setdb1* KO: $n = 3044$). TADs structures were largely maintained with both directionality index scores and defined boundaries being conserved (Fig. 4c, d and Supplementary Fig. 4b). Collectively, H3K9me3 depletion did not lead to an extensive perturbation of chromatin organization at the subnuclear compartment or TAD levels, and transcriptional dysregulation was, for the most part, not associated with changes in domain and compartment structures.

## Aberrant CTCF and SMC3 binding are associated with disrupted chromatin loops and altered transcription

CTCF is an architectural protein with versatile roles in genome organization[39]. In the loop extrusion model, CTCF functions in concert with the cohesin complex to mediate chromatin loop formations[27]. These loops cooperatively direct the *cis*-regulatory network for genes[57]. To appreciate the consequences of the aberrant CTCF binding in *Setdb1* KO ESCs, we integrated CTCF and SMC3 ChIP-seq datasets with Hi-C interactions and chromatin loop definitions. Increased SMC3 enrichment was similarly detected at increased CTCF peaks in the *Setdb1* KO cells (Supplementary Fig. 2f). To determine whether 3D organization was impacted, we applied HiCCUPS to define chromatin loops (WT: $n = 10,122$; *Setdb1* KO: $n = 10,598$), with CTCF peaks found at 76.4% of all loop anchors. Relative to WT, upon *Setdb1* deletion, we found 794 and 499 loops with increased and decreased interactions, respectively (Supplementary Fig. 5a). We detected significantly more increased CTCF peaks ($n = 201$) than number-matched, randomly selected CTCF peaks ($n = 63$) localized at increased loop anchors. Notably, such enrichment was not observed at decreased or unchanged loops (Fig. 5a, b and Supplementary Fig. 5b). We discovered that the increased loops disrupted pre-existing structures, leading to weakening of the other loops (Fig. 5a). It is worth noting that most altered loops (96.2%) had changed interactions within said loop, and not only at the anchors.

As chromatin loops could affect the interactions between distal *cis*-regulatory elements, we asked if transcription was affected within altered loops. Downregulated genes in *Setdb1* KO were located within decreased loops (62/286) at a significantly higher incidence than randomly selected genes (Fig. 5c and Supplementary Fig. 5c). Moreover, the majority of the upregulated genes in *Setdb1* KO were not marked by H3K9me3 in the WT cells (574/660), a subset of which were located in increased loops (Supplementary Data file 2). Therefore, we postulated that the expression of these genes could be modified by 3D interaction changes. It should be noted that although dysregulated genes and repeats showed a subtle but significant occurrence in altered compared to random chromatin loops, not all dysregulated transcripts were associated with altered chromatin loops (Fig. 5d and Supplementary Fig. 5d). This indicated that while changes in higher-order chromatin interactions could impact transcription, it was not a prerequisite.

A recent study reported that H3K9me3 and H3K36me3 concomitantly marked poised enhancers and were co-regulated by SETDB1 and NSD proteins[12]. Integrating publicly available H3K36me3 ChIP-seq datasets[12], we found that the H3K9me3/H3K36me3 co-enriched dual domains and the H3K9me3-regulated CTCF binding sites are largely distinct (Supplementary Fig. 5e). In fact, we do not detect substantial H3K36me3 ChIP-seq signal at SETDB1-regulated CTCF peaks in WT or *Setdb1* KO cells (Supplementary Fig. 5f). Moreover, genes upregulated in SETDB1 depleted cells do not show increased 3D chromatin interactions with the H3K9me3/H3K36me3

dual domains (424/440) (Supplementary Fig. 5g). These results demonstrate that our observations on CTCF binding modulation are not associated with H3K36me3 or NSDs.

To further evaluate consequences on long-range *cis*-interactions, we employed FitHiC to identify significant interactions in WT and *Setdb1* KO ESCs ($q < 0.1$) ($n = 4,293,766$). Focusing on the interactome of upregulated genes and repeats that are not marked by H3K9me3, we discovered that a subset showed increased interaction frequencies with loci that exhibited loss of H3K9me3 in *Setdb1* KO cells, concomitant with increased active epigenetic features (Supplementary Fig. 5g). For example, the *Ccny* gene, which harboured no H3K9me3 at its promoter in WT cells, was significantly upregulated upon SETDB1 depletion. No changes in H3K27ac, chromatin accessibility, and DNA methylation levels were detected (Supplementary Fig. 5h). Furthermore, the *Ccny* promoter gained interactions with a distal bin, likely through the enhanced CTCF and cohesin binding. This distal region was highly accessible and showed substantial increase of H3K27ac and depletion of H3K9me3 enrichment levels in *Setdb1* KO cells (Supplementary Fig. 5i). Overall, this subset of upregulated genes and repeats, which were not directly regulated by H3K9me3, gained interactions with SETDB1 targets that harboured putative enhancers signatures. On the other hand, change in chromatin landscapes could also underlie transcriptional downregulation. For instance, the *Vegfa* gene was downregulated in *Setdb1* KO cells and resided in decreased chromatin loops. Intriguingly, we detected concomitant loss of interactions with a locus harbouring active *cis*-regulatory marks and gained interactions with another highly DNA methylated region (Supplementary Fig. 6). Taken together, these data revealed that the aberrant CTCF binding in *Setdb1* KO cells was associated with altered chromatin loop structures. The change in chromatin landscapes within said loops could result in differential interactions, leading to transcriptional dysregulation.

The loss of *Setdb1* in ESCs has been shown to be associated with a higher propensity for differentiation towards the trophectoderm (TE) lineage[5,58,59]. Consistent with previous observations, *Cdx2*, which was not marked by H3K9me3 in the WT cells, was upregulated in *Setdb1* KO ESCs (Supplementary Fig. 5j). This TF is required for TE specification and promotes expression of TE genes, while also repressing pluripotency genes[60]. Consistent with prior findings, we observed that *Cdx2* downstream targets, including *Tfap2c* and *Eomes*, were similarly upregulated. However, the underlying mechanism of how SETDB1 mediates its silencing is unclear. Interestingly, we found that the *Cdx2* gene resided in an increased loop anchor and significantly interacted more frequently with a distal bin (labelled bin B) with increased CTCF binding (Fig. 5a). Remarkably, we detected no change of CTCF binding or of *Cdx2* expression in the *Dnmt* TKO cells (Supplementary Fig. 5j, k), suggesting that the CTCF-mediated interaction with bin B is associated with increased *Cdx2* transcription. Furthermore, we analyzed CTCF ChIP-seq and RNA-seq data derived from WT and *Setdb1* KO neurons[34] and did not observe such alterations (Supplementary Fig. 5j, k). While increased CTCF peaks were also defined in SETDB1-depleted neurons, the majority of them were distinct from the set defined in *Setdb1* KO ESCs (Supplementary Fig. 5l, m). Notably, we found that most sites with significantly increased CTCF binding possessed motifs that are classified as "low occupancy sequences" (Supplementary Fig. 2g). In previous reports, CTCF binding sites were categorized into low-, mid-, and high occupancy (LowOc, MidOc, and HighOc), based on the degree of CTCF ChIP-seq signal at the given locus[61,62]. LowOC sites were highly associated with developmentally regulated CTCF binding. This is in line with the cell-type differential targeting of SETDB1, which determines CTCF binding at given loci in a tissue-specific manner. Indeed, out of all analyzed P0 samples, *Cdx2* is only expressed in the intestine tissues and show concordant CTCF binding at both *Cdx2* and bin B (Supplementary Fig. 5n), analogous to *Setdb1* KO ESCs. Furthermore, we compared WT and *Setdb1* KO ESCs and neurons and found that SETDB1 prevented cell-type differential CTCF binding. For instance, in

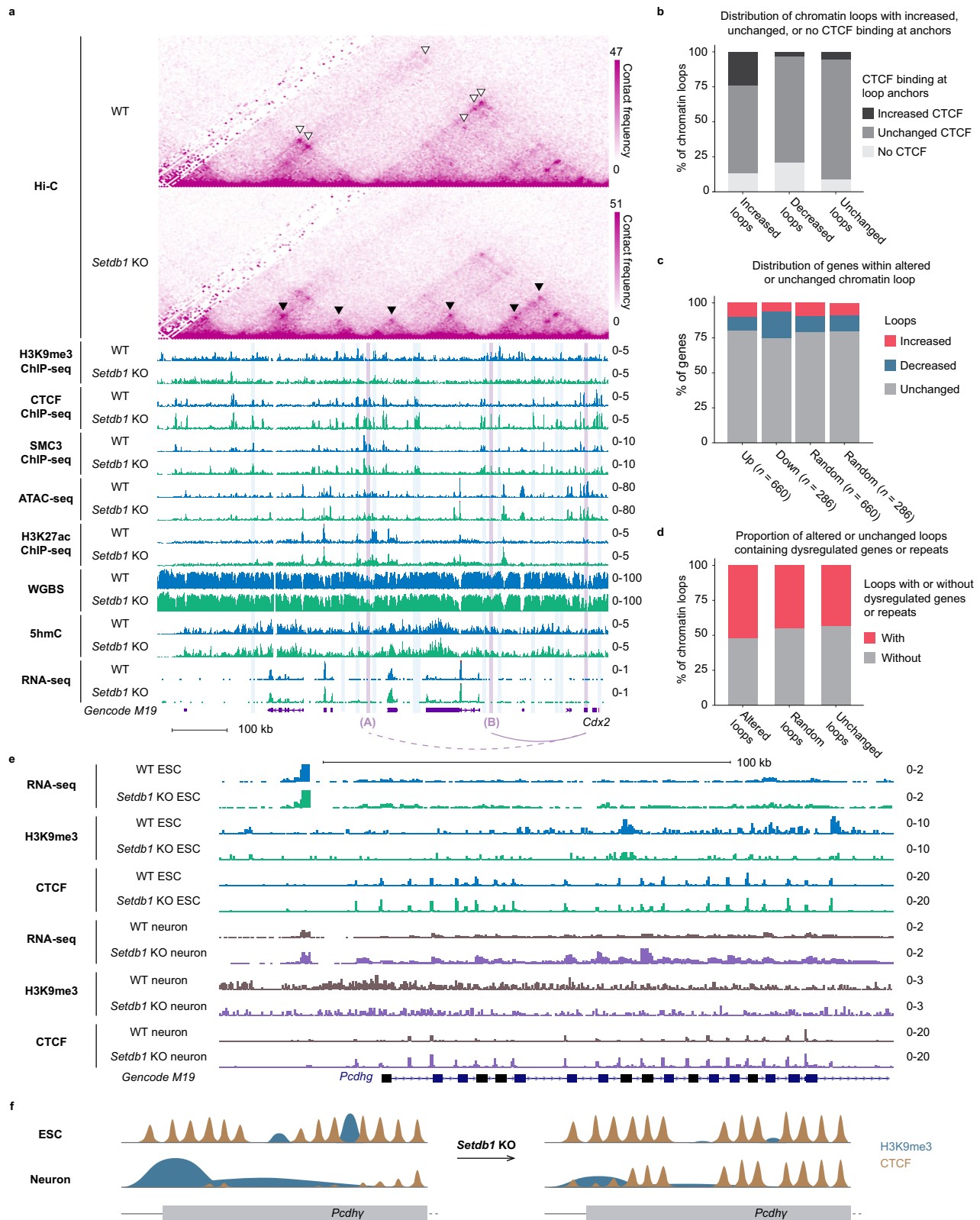

neurons, SETDB1 represses particular CTCF sites at the protocadherin gene cluster[34]. These same sequences are bound by CTCF in WT ESCs (Fig. 5e, f). Conversely, we also found in ESCs, H3K9me3 silences loci that are normally CTCF bound in neurons (Supplementary Fig. 5o). Taken together, our results revealed that cell-type specific H3K9me3 plays a role in regulating CTCF binding and higher-order chromatin structures, which may influence cellular identities.

## Discussion

SETDB1 is an important H3K9 methyltransferase that is necessary for proper developmental progression and essential cellular processes in mice. Many studies have described the transcriptional repression of genes and repetitive elements mediated by this enzyme[8,12,15–17,34,63]. However, while collapse of a specific TAD in SETDB1-deficient neurons has been reported[34], how H3K9me3 and SETDB1 influence the 3D

**Fig. 5 | Chromatin loops are disrupted in *Setdb1* KO cells and a subset are associated altered transcription. a** A genome browser screenshot demonstrates increased (black arrowheads) and decreased loops (white arrowheads), and increased CTCF binding (blue shading) on chromosome 5 (chr5:146,525,455-147,352,727) upon *Setdb1* KO. The non-H3K9me3 marked upregulated gene (*Cdx2*) demonstrates the association between changes in chromatin structure and aberrant gene expressions. *Cdx2* significantly lost (dashed arc) and gained (solid arc) interactions with two distal bins in *Setdb1* KO, loci A and B (purple shading), respectively. Hi-C contact heatmaps display the raw counts in 5 kb windows. H3K9me3, CTCF, SMC3, H3K27ac, and 5hmC tracks are displayed as input subtracted RPKM values. ATAC-seq tracks are displayed as normalized read counts. WGBS datasets are shown as mCG/CG ratio for individual CpG sites with coverage ≥ 5 reads. RNA-seq datasets are shown as RPM. **b** Stacked bar chart shows the distribution of chromatin loops that contain increased, unchanged, or no CTCF binding at the anchors. Loops are classified as increased, decreased, and unchanged loops. **c** Stacked bar chart illustrates the distribution of up- and down-regulated genes or number-matched randomly selected genes that fall into increased, decreased, and unchanged loops, respectively. The randomly selected genes were picked twice independently ($n = 660$ and $n = 286$), corresponding to the numbers of up- and down-regulated genes, respectively. **d**) Stacked bar chart illustrates proportion of altered loops, number-matched randomly selected loops ($n = 1293$) and unchanged loops that contain dysregulated genes or repeats. **e** A genome browser screenshot demonstrates SETDB1 represses CTCF occupancy in neurons but not in ESCs at *Pcdhy* gene cluster. Gain of CTCF binding is only observed in neurons after SETDB1 depletion, accompanied by upregulation of the *Pcdhy* genes. RNA-seq datasets are shown as RPM values. H3K9me3 and CTCF ChIP-seq tracks are displayed as input subtracted RPKM values. **f** A schematic illustrates the cell-type specific regulation of CTCF binding by SETDB1. The CTCF sites at *Pcdhy* locus, which are normally utilized in ESCs, are repressed by SETDB1/H3K9me3 in neurons. Upon *Setdb1* deletion, while ESCs, show subtle changes, neurons show a substantial gain of CTCF binding concomitant with reduced H3K9me3 levels. Source data are provided as a Source Data file.

genome architecture has not been fully explored. Here, we demonstrated a global mutual exclusivity of H3K9me3 enrichment and CTCF binding across mouse tissues. Between cell/tissue-types, we found a subset of loci with dynamic H3K9me3, which harbored CTCF motifs. In one cell-type, H3K9me3 can block CTCF, while in another, the absence of the modification at the same locus allows CTCF to bind. Given that epigenomic reprogramming and reconfiguration of chromatin architecture are essential during differentiation and development[53,64–67], our results suggest that differential H3K9me3 regulates cell/tissue-type specific CTCF binding and describe an addition layer of establishing different chromatin structures between cellular identities.

To investigate consequences of SETDB1 loss, we examined epigenetic modifications, chromatin accessibility, nucleosome occupancy, and transcriptomic changes in WT and *Setdb1* KO ESCs. In line with the model of SETDB1 shielding unused CTCF binding sites, we found global gains of CTCF enrichment[34]. Given that H3K9me3 enrichment is highly correlated 5mC levels, we sought to separate the role of the two epigenetic marks. We discovered that the sole loss of 5mC in *Dnmt* TKO cells was insufficient to cause the increased CTCF binding at these sites. Moreover, H3K9me3 enrichment was maintained in *Dnmt* mutants. It should be noted that consistent with the known binding properties of CTCF, we also observed sites with increased CTCF ChIP-seq signal in *Dnmt* TKO cells. However, these were not the same sites as those regulated by SETDB1. As previously reported, due to cell lethality of *Setdb1* KO ESCs, the subtle changes in the DNA methylome could be a result of the brief available window for measurement[63]. Therefore, it remains possible that the overlap of increased CTCF sites in SETDB1 and DNMTs depleted ESCs could potentially be higher than observed. Nevertheless, our results indicate that at a subset of loci, SETDB1 predominates in preventing CTCF binding. It should be noted that these sites are also distinct to those regulated by G9a-mediated H3K9me2, which were previously reported[40]. Interestingly, motif analysis of the SETDB1-repressed CTCF binding sites revealed enrichment of some sequence variants, including the replacement of CpG dinucleotides at two known targets of DNA methylation. These changes were also more common among the tissue-specific LowOC CTCF binding sites. It is possible that the lower occurrence of these CpG sites renders DNA methylation less effective at repression. Hence, H3K9me3 predominates as a parallel pathway; however, further investigation is needed to prove this hypothesis.

Mechanistically, we confirmed depletion of H3K9me3 led to reorganization of nucleosomes. The nucleosome depletion and increased chromatin accessibility allow for CTCF to bind to chromatin. Many of these sequences were located within retrotransposons. Surprisingly, in addition to the class I and II ERVs, SINE B2 elements were also identified as a target of SETDB1. As these elements are highly enriched with CTCF motifs, rather than transcriptional or *cis*-regulatory silencing, SETDB1 represses them by antagonizing CTCF

binding. Moreover, we detected significantly higher binding of KAP-1 at these SINE B2 elements in WT ESCs. Its association with specific KRAB-ZNF proteins is likely the method for SETDB1 recruitment to such targets. We suggest that this mechanism functions as the way that different cell-types can selectively utilize distinct CTCF sites to regulate chromatin loops, which in turn control transcriptional networks appropriate for that cellular identity.

While the relationships between H3K9me3 and transcriptional repression has been long established[1–4], their connection to chromatin architecture remains relatively underappreciated. B compartments are highly associated with nuclear periphery localization and transcriptionally inert heterochromatin marked by H3K9me3. During development and cellular differentiation, repressed genes that are activated, undergo B-to-A compartment switching, with some leading to altered TAD boundaries and replication timing[53,56,68,69]. Moreover, a recent study demonstrated that DNMT inhibitor treatment induced transcriptional upregulation that was coupled with B-to-A compartment switching and altered domain boundaries[56]. Consistent with these reports, we observed the association of H3K9me3 enrichment and B compartments in WT ESCs. Surprisingly, our Hi-C analysis revealed that compartmentalization and TADs largely remained unchanged after SETDB1 depletion. Upregulated transcripts showed higher concordance with H3K9me3 loss than B-to-A compartment switching. While some genes and repetitive elements demonstrated concomitant changes in expression and compartment scores, the majority of differential transcription was not accompanied by altered chromatin organization at 25 kb resolution. It was previously shown that in mouse ESCs, DNA methylation and H3K9me3 silence largely distinct targets[16]. Therefore, the loci that are derepressed and switch from B-to-A compartments upon DNMT inhibition[56] are likely different from the ones that are upregulated in *Setdb1* KO cells. Collectively, our data suggests that transcriptional change of different loci affects chromatin structures in diverse ways and on a genome-wide level, transcription is not sufficient to induce subnuclear compartment switching and domain changes.

Many of the dysregulated genes in *Setdb1* KO cells are not marked by H3K9me3, leading to previous studies to classify them as downstream or indirect effects[16]. Interestingly, our Hi-C data revealed that the formation of aberrant chromatin loops in *Setdb1* KO cells could impact transcription via modulating CTCF binding. For example, the *Cdx2* gene, which encodes for a key TE regulator, gained interactions with a distal locus that is normally marked by H3K9me3. The aberrant chromatin loops in *Setdb1* KO ESCs brings the gene into close spatial proximity to a putative *cis*-regulatory element, which is concomitant with its increased expression. Notably, these changes in *cis*-regulatory interactions are not associated with the dysregulation of H3K9me3/H3K36me3 marked poised enhancers. Consistent with known functions of CDX2, we observed the upregulation of TE genes and downregulation of pluripotency genes. This includes *Nanog*, an established marker gene of

pluripotency[70,71], which is downregulated upon depletion of SETDB1[5,58]. Interestingly, our data reveals that *Nanog* also resides within a chromatin loop, which is disrupted in *Setdb1* KO cells and is associated with aberrant CTCF binding (Supplementary Data file 2). The alteration is accompanied by differential epigenetic signatures of several interacting loci. Therefore, it is possible that decreased levels of *Nanog* arises in parallel to CDX2-mediated repression. It is worth noting that, the transcriptional change of *Cdx2* and *Nanog* occurs independently of DNA hypomethylation, as the expression of both are not significantly different in the *Dnmt* TKO as compared to WT ESCs.

Another recent study reported the presence of "domains involved SETDB1 and cohesin" (DiSCs), which presumably affects transcription and genome topology in a H3K9me3-independent manner[72]. Our results from ESCs that harbor a catalytically inactive SETDB1 revealed that the regulation of CTCF binding is indeed H3K9me3 dependent. While our CTCF and cohesin binding profiles are similar to those described in several published studies, they were largely distinct from the cohesin and SETDB1 ChIP-seq datasets from WT and *Setdb1* KD E14 ESCs generated by Warrier and colleagues[72] (Supplementary Fig. 5o). This difference could underly the disagreement in our models of how SETDB1 regulates nuclear architecture. While this manuscript was under preparation, Gualdrini et al. reported similar findings, demonstrating that H3K9me3 restricts CTCF recruitment at SINE B2 elements in human macrophages[73]. With SETDB1 depletion, formation of new chromatin interactions was associated with transcriptional upregulation of specific genes. These results align with our observations, which offer additional insights into the underlying molecular mechanism. This includes the alteration in cohesin binding, the redistribution of nucleosomes, and how this phenomenon functions by the catalytic activity of SETDB1 and is independent of DNA methylation. Taken together, in both human and mouse, SETDB1 appears to play cell-type specific roles in regulating CTCF binding.

In summary, we propose a mechanism, where H3K9me3 controls transcription by regulating the long-range interactions between genes and distal enhancers. The disruption of existing chromatin landscape could lead to altered expression, regardless of H3K9me3 enrichment at the gene or its *cis*-regulatory elements. Conversely, other genes are upregulated due to increased interactions with enhancers or derepression of elements that are already in close spatial proximity (Fig. 6). Some of the previously defined downstream genes are indeed regulated directly by SETDB1, but not through H3K9me3-mediated repression of their promoters. This model could in part explain the distinct functions of SETDB1 across developmental stages and cell-types. A proportion of the tissue-specific H3K9me3 is responsible for regulating differential CTCF-mediated chromatin loops. In addition to its critical role in development, SETDB1 has also been classified as an oncogene, with its amplification reported in many human cancers[74–76]. Insights into its impact on cell-type specific nuclear organization and its mechanisms to influence transcription may provide new therapeutic possibilities.

## Methods
### Cell culture
WT mouse ESCs (clone 33#6) and two ESC lines carrying a SETDB1 catalytic mutant (clone 33#6$^{CI243A+}$#21 and 33#6$^{CI243A+}$#31)[8] were maintained in ES medium (DMEM with 15% certified FBS (Life Technologies), 20 mM HEPES, 0.1 mM non-essential amino acids, 0.1 mM 2-mercaptoethanol, 100 U/ml penicillin, 0.05 mM streptomycin, 1 U/ml Leukemia-inhibitory factor (LIF), and 2 mM glutamine). Cells were passaged at 80% confluence. *Setdb1* conditional KO was induced by the addition of 800 nM (Z)−4-Hydroxytamoxifen (4-OHT) (Abcam, 141943) for 4 days and harvested on day 6 post 4-OHT treatment.

### Western blot
Whole cell lysates were extracted from $1 \times 10^7$ ESCs using radioimmunoprecipitation assay buffer (RIPA buffer) with 1X PIC and

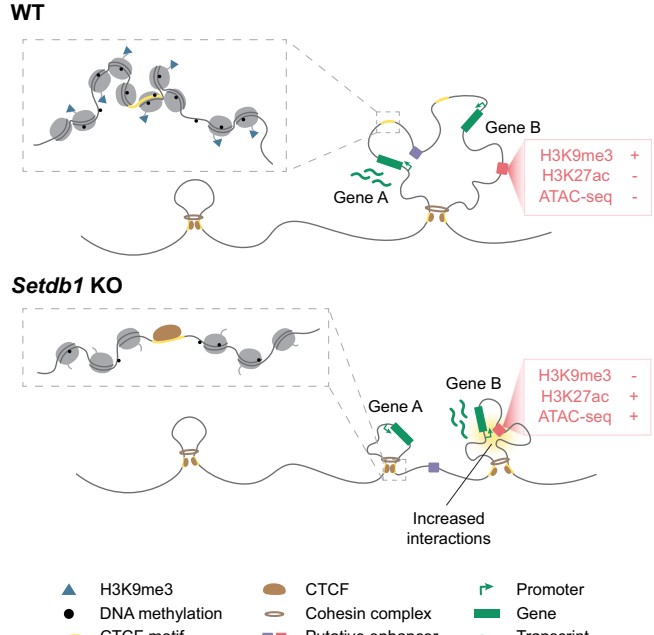

**Fig. 6 | Model of SETDB1 preventing aberrant CTCF binding and regulating 3D chromatin interactions.** Our proposed model of H3K9me3-dependent regulation of CTCF and 3D chromatin interactions. In WT cells (top), a proportion of CTCF motifs are marked by H3K9me3, which leads to closed chromatin states. This prevents CTCF binding and occurs independently of DNA methylation. In *Setdb1* KO cells (bottom), loss of H3K9me3 allows increased binding and redistribution of CTCF and cohesin, which could lead to altered loop and disruption of pre-existing structures, leading to transcriptional downregulation (gene A). Moreover, changed chromatin environments can aberrantly bring putative enhancers into proximity of genes and repeats to alter transcription (gene B).

phosphatase inhibitor phosSTOP™ (Roche). Protein concentrations were quantified using Quick Start™ Bradford Protein Assay (Bio-Rad) according to the manufacturer protocol. Lysates were subsequently resolved by 15% SDS-PAGE gel, wet-transferred onto PVDF membranes, and probed with 1:1000 anti-SETDB1 (ThermoFisher, PA5-30334), 1:1000 anti-CTCF (Active Motif, 61311), 1:1000 anti-FLAG (Sigma-Aldrich, F3165) or 1:20000 anti- β-Tubulin (Abcam, 6046) antibodies. Signals were developed with Clarity™ Western ECL Substrate (Bio-Rad) according to manufacturer's protocol. Images were detected with ChemiDoc™ Touch Imaging System (Bio-Rad).

### Dot blot
MODified Histone Peptide Array (Active Motif) was prepared according to manufacturer's instructions. In brief, the array slides were incubated for 2 hours in blocking buffer (5% non-fat milk in TBST) at room temperature and rinsed 3 times with TBST. The array slides were then incubated with primary antibody (H3K9me3 Abcam, 8898) at a dilution of 1:5000 overnight at 4 °C, washed and then incubated in HRP-conjugated secondary antibody for 2 hours at room temperature. Chemiluminescence image of the array slides were detected and captured using ChemiDoc™ Imaging System (Bio-Rad). Images were analysed using Array Analyze Software (Active Motif) to determine the specificity of the antibody using automatic background level. Multiple peptide average is used to display an overview of the antibody's specificity.

### Native and Crosslink Chromatin Immunoprecipitation-sequencing (ChIP-seq)
Native ChIP was performed with $1 \times 10^7$ of cells. Cells were resuspended in cold douncing buffer (10 mM Tris-Cl pH7.5, 4 mM MgCl$_2$, 1 mM

CaCl$_2$, 1X PIC) and homogenized 20 times with 25 gauge 5/8 inch needle. 0.8 uL of MNase (Thermo Scientific™, 88216) was added to digest chromatin at 37 °C for 10 min. 10 mM EDTA was added to the homogenate to quench the digestion on ice for 5 min. Cells were incubated with hypotonic lysis buffer (0.2 mM EDTA pH8.0, 0.1 mM benzamidine, 0.1 mM PMSF, 1.5 mM DTT, 1X PIC) on ice for 1 hour and vortexed every 10 min. Lysate was centrifuged to remove debris and incubated overnight with prewashed Dynabeads™ M-280 Sheep Anti-Mouse IgG (Invitrogen™, 11202D) that bound with rabbit anti-H3K9me3 (Abcam, 8898) or anti-H3K27ac (Active Motif, 39133) antibodies. The beads were washed twice with ChIP wash buffer (20 mM Tris-Cl pH8.0, 0.10% SDS, 1% Triton X-100, 2 mM EDTA, 150 mM NaCl, 1X PIC) and once with final ChIP wash buffer (20 mM Tris-Cl pH8.0, 0.10% SDS, 1% Triton X-100, 2 mM EDTA, 500 mM NaCl, 1X PIC). Subsequently, beads were eluted in elution buffer (100 mM NaHCO$_3$, 1% SDS) and treated with ribonuclease (RNase A) and Proteinase K at 68 °C for 2 hours. Samples were purified with QIAquick PCR Purification Kit (Qiagen) according to manufacturer protocol.

Crosslink ChIP was performed with $1 \times 10^7$ of cells. Cells were crosslinked in 1% formaldehyde for 10 min and were resuspended in cold lysis buffer (50 mM Tris-Cl pH8.0, 1% SDS, 10 mM EDTA, 1 mM PMSF, 1X PIC). Chromatin was sonicated (Covaris S220 Focused-ultrasonicator) for 450 s at 4 °C at 175 W with 10% of duty factor and 200 cycles per burst and maintained on ice. Sonicated chromatin was centrifuged to remove debris and diluted 1:4 in IP dilution buffer (20 mM Tris-Cl pH8.0, 0.15 M NaCl, 2 mM EDTA, 1% Triton X-100, 1X PIC). Immunoprecipitation was performed using with Dynabeads™ Protein A (Invitrogen™) with 5 µg of rabbit anti-CTCF (Active Motif, 61311) or anti-SMC3 antibody (Abcam, 9263) at 4 °C for 2 hours. Beads were washed with wash buffer 1 (20 mM Tris-Cl pH8.0, 0.15 M NaCl, 2 mM EDTA, 1% Triton X-100, 0.1% SDS, 1 mM PMSF), then with wash buffer 2 (20 mM Tris-Cl pH8.0, 0.5 M NaCl, 2 mM EDTA, 1% Triton X-100, 0.1% SDS, 1 mM PMSF), then with wash buffer 3 (10 mM Tris-Cl pH8.0, 0.25 M LiCl, 1 mM EDTA, 0.5% IGEPAL CA-630, 0.5% DOC), and finally with 1X TE (10 mM Tris-Cl pH8.0, 1 mM EDTA). Subsequently, chromatin was eluted from beads in elution buffer (25 mM Tris-Cl pH7.5, 5 mM EDTA, 0.5% SDS) and treated with ribonuclease (RNase A) and Proteinase K at 37 °C for 1 hour followed by 65 °C overnight. Samples were purified with QIAquick PCR Purification Kit (Qiagen) according to manufacturer protocol.

All ChIP-seq libraries were prepared according to the Illumina protocol and sequenced on the Illumina NextSeq 500 platform.

### Assay for Transposase Accessible Chromatin with high-throughput sequencing (ATAC-seq)

ATAC-seq protocol was adapted from Buenrostro et al. [77] with minor modifications. $5 \times 10^4$ fresh cells were lysed in cold lysis buffer (10 mM Tris-Cl pH7.4, 10 mM NaCl, 3 mM MgCl$_2$, 0.1% IGEPAL CA-630, 0.1% Tween-20, 0.01% Digitonin (Invitrogen™ BN2006), 1X PIC) for 3 min. Cells were collected by centrifugation at 4 °C for 10 min at 500 g. Transposition reaction mix containing Tn5 (Vazyme) were added to the cells and incubated at 37 °C for 30 min. Tagmented DNA was purified with MiniElute PCR Purification Kit (Qiagen) and amplified using KAPA 2X HiFi Hotstart Mix (KAPA Biosystem) with corresponding Nextera primers (Illumina). Amplified DNA was purified and size selected using AMPure XP beads (Beckman and Coulter). Libraries were sequenced on the Illumina NextSeq 500 platform.

### Micrococcal nuclease digestion with deep sequencing (MNase-seq)

MNase-seq protocol was adapted from Cui and Zhao et al. [78] with minor modifications. $2 \times 10^6$ cells were homogenize with 25 gauge 5/8 inch needle, in cold homogenization buffer (10 mM Tris-HCl pH7.5, 4 mM MgCl$_2$, 1 mM CaCl$_2$, 1X PIC). 0.8uL of MNase (Thermo

Scientific™, 88216) was added to the homogenate and incubated at 37 °C for 2 min. 10 mM EDTA was added to the homogenate to quench the digestion on ice for 5 min. Cold hypotonic Lysis buffer (0.2 mM EDTA, 0.1 mM Benzamidine, 0.1 mM PMSF, 1.5 mM DTT, 1X PIC) was added to the reaction mixture and incubated on ice for 1 hour. The pellet was centrifuged at 3000 g for 5 min and supernatant was collected. DNA was recovered using QIAquick PCR Purification Kit (Qiagen) following manufacture instruction. Libraries were built using KAPA HyperPrep kit according to manufacturer protocol and converted to DNBSEQ-G400 compatible library using MGIEasy Universal Library Conversion Kit (App-A) (MGI tech, 1000004155) following manufacturer instructions. The libraries were sequenced on the MGI Tech DNBSEQ-G400 sequencer.

### Hi-C
Hi-C was performed using Arima-HiC Kit (Arima Genomics) according to manufacturer protocol. $1 \times 10^6$ of cells were used as starting material. DNA fragmentation was done with the same setting as in ChIP on the Covaris S220 Focused-ultrasonicator. Libraries were built using KAPA Hyper Prep kit according to manufacturer protocol. Two biological replicates were prepared for each sample. The libraries were sequenced on the Illumina Novaseq sequencer.

### Bioinformatic analysis
**ChIP-seq and 5hmC Capture-seq data analysis.** Reads were aligned to the mm10 (GRCm38) mouse reference genome, using Bowtie v1.2 with parameters -v 3 -m 1 --best --strata. Duplicates were removed with Picard MarkDuplicates v2.9.0. For ChIP-seq data, input subtracted reads per kilobase per million reads (RPKM) values were calculated for 100 bp bins[63]. For 5hmC Capture-seq data, RPKM values were calculated per 100 bp bins.

Peaks for H3K9me3 were called by EPIC v0.2.9 with parameters -k -w 200 -g 3 -fs 200 -egf 0.8917. Peaks for CTCF were called by MACS2 v2.1.0 with parameters --keep-dup all -g 2.43e9 -q 0.05. Motif analysis for H3K9me3 peaks was performed by HOMER with the whole genome as background[79]. Poisson $p$ value of pairwise comparison of ChIP data was calculated. Adjusted $p$ value was calculated with Benjamini-Hochberg method. Increased CTCF peaks were defined as significantly increased CTCF enrichment ($q < 5e-4$ in *Setdb1* KO ($n = 2669$), and $q < 5e-2$ in *Dnmt* TKO ESC ($n = 2259$)).

**RNA-seq data analysis.** Reads were aligned to the mm10 (GRCm38) mouse reference genome and transcriptome assembly GENCODE GTF vM15 separately with the Spliced Transcripts Alignment to a Reference (STAR) v2.5.3a[80]. The uniquely aligned reads to the genome were used for further analysis. Gene expression values were quantified by fragments per kilobase per million reads (FPKM) using RSEM[81]. Expression values of repetitive elements were quantified by uniquely aligned read counts.

Poisson $p$ value of pairwise comparison of expression data was calculated as previously described[40]. $p$ values were adjusted by Benjamini-Hochberg method. Dysregulated genes were defined as $q < 0.001$ and fold change > 2. Dysregulated individual repetitive elements were defined as $q < 0.01$ and fold change > 2.

**Whole Genome Bisulfite Sequencing (WGBS) data analysis.** Reads were aligned to the mm10 (GRCm38) mouse reference genome by Bismark v0.29.0[82]. Duplicates were removed with Picard MarkDuplicates v2.9.0. To determine the efficiency of bisulfite conversion, reads were mapped to lambda genome. To calculate the mean methylated CpG percent in regions of interest, window size of 100 bp was used, with those having less than 5 aligned reads being filtered out. For visualization, individual CG methylation levels were extracted for CG sites with coverage of ≥ 5 reads.

**ATAC-seq data analysis.** Low quality reads (Phred, score ≤20) were removed and adaptors were trimmed (Trim Galore v0.4.3). Reads were aligned to the mm10 (GRCm38) mouse reference genome (Bowtie2 v2.3.3.1)[83]. Reads with more than one best alignment and duplicates were removed (Picard MarkDuplicates v2.9.0). The alignments were then adjusted to set the start site of each read as the center of the transposase binding (MACS2 v2.1.0)[77,84]. Reads Per Million (RPM) normalized signal were generated for visualization. Peaks were called by MACS2 with parameters --nomodel --keep-dup all -p 0.01 to define open chromatin sites. Read counts were quantified by GenomicAlignments v1.18.1[85]. To make sure the signal to noise ratio uniform, we conducted signal normalization by S3norm with default parameters (bin size of 200 bp)[86].

**MNase-seq data analysis.** Adapter sequence contamination of the raw reads of each MNase-seq libraries were removed by Trim Galore v0.6.6 with parameters --paired -a AAGTCGGAGGCCAAGCGGTCTT AGGAAGACAA -a2 AAGTCGGATCGTAGCCATGTCGTTCTGTGAGCCA AGGAGTTG. Trimmed reads were mapped to the mm10 (GRCm38) mouse reference genome using Bowtie2 v2.4.2 with parameters -q --phred33 --end-to-end --sensitive --no-mixed --no-discordant. PCR duplicates were excluded in downstream analysis using Picard MarkDuplicates v2.23.8. Mapped read pairs with less than 4 mismatches were retained using bamtools v2.5.1 with parameter filter -tag "XM:< 4". Based on the inferred fragment size of each pair, fragments with length between 141 to 190 bp were passed to DANPOS v2.2.2 with parameter dpos -m 1 to define the location of nucleosome. Nucleosome occupancy was quantile normalized with WIQ function in DANPOS and visualized in with deepTools v3.0.1.

**Hi-C data analysis.** The alignment was processed by Juicer v1.5.6[87] mapped to mouse reference genome mm10 (GRCm38). All the other parameters of Juicer were set to default. Hi-C quality metric can be found in Supplementary Table 1. Replicates were merged for subsequent analysis.

A/B compartments were called by cooltools v0.3.2[88] call-compartments function at 25 kb resolutions. The first three eigenvectors were calculated for each bin. The eigenvalues that best correlated with ATAC-seq signals (RPM) were reported as compartmentalization score. The sign of eigenvalues was adjusted per chromosome to positively correlate with ATAC signal. Hence, A and B compartments were represented by positive and negative values, respectively.

Hi-C contact matrices with Knight-Ruiz (KR) normalization[89] were generated by Juicer dump and used in producing directionality index and TADs calls by Hi-C Domain Caller[20] at 25 kb resolution.

**Hi-C chromatin loop analysis.** Interaction frequency of chromatin loops called at 10 kb and 5 kb resolution by Juicer HiCCUPS with the default setting for high resolution maps[18] (WT: $n = 10,122$, *Setdb1* KO: $n = 10,598$). Loops called from different samples with exactly the same anchors were merged to keep only one set of definitions, generating a combined set of chromatin loops ($n = 16,554$). Poisson test was conducted to the normalized contacts counts for each chromatin loop between *Setdb1* KO versus WT. The details of the test and normalization method were the same for significant interaction analysis. Increased and decreased chromatin loops were defined as $q < 0.1$, fold change > 2 ($n = 794$) and $q < 0.1$, fold change <0.5 ($n = 499$), respectively.

**Hi-C significant interactions.** Significant interactions were called by Fit-Hi-C at 25 kb, 10 kb, and 5 kb resolutions[90]. Interactions with $q < 0.1$ in any of the samples were kept. For significant interactions completely overlapped at different resolutions, only the highest resolution was kept. All merged significant interactions constitute the set of "all significant interactions" ($n = 4,293,766$) used in further analysis.

For each sample, the counts of the significant interactions were scaled down to the smallest Hi-C contacts number and the differential test was conducted for *Setdb1* KO versus WT as previously described[40]. $p$ values were adjusted by the Benjamini-Hochberg method. Increased and decreased significant interactions were defined as $q < 0.1$, fold change > 2 ($n = 16,087$) and $q < 0.1$, fold change <0.5 ($n = 7926$), respectively.

**Enrichment analysis.** Enrichment analysis was conducted (shown in Supplementary Fig. 5b–d). Random distribution was obtained by density function from 10,000 times random simulation. $p$ values are obtained by non-parametric bootstrapping.

**Reporting summary**
Further information on research design is available in the Nature Portfolio Reporting Summary linked to this article.

## Data availability
All sequencing datasets generated in this study have been deposited under the accession number GEO: GSE184471. Source data are provided with this paper. Sources of publicly available NGS Datasets used in integrative analysis are included in Supplementary Data file 3. Source data are provided with this paper.

## Code availability
All code used in this study was previously published and no customized code was used in this manuscript.

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

## Acknowledgements

This work is supported by the Hong Kong Research Grant Council (GRF16102817 and 16103423), the Croucher Innovation Award and the Hong Kong Epigenome Project (Lo Ka Chung Charitable Foundation) to D.L. We thank Dr. Matthew Lorincz for the ESC lines and the staff at the HKUST Center for Epigenomics Research and BioCRF for their assistance.

## Author contributions

P.L.F.T. and D.L. designed the study. P.L.F.T., L.Y.C., and M.F.C. performed all experiments. P.L.F.T. and M.F.C. performed all data analyses. P.L.F.T. and D.L. wrote the manuscript.

## Competing interests

The authors declare no competing interests.
