## [Peer Review File · Nature Communications]

Cell-type differential targeting of SETDB1 prevents aberrant CTCF binding, chromatin looping, and cis-regulatory interactionsREVIEWER COMMENTS

Reviewer #1 (Remarks to the Author):

Reviewer #2:

The reviewer is satisfied with the major responses of the authors, but would like the following minor points to be corrected. Once this correction is made, there will be no further comments.

5. Fig. 5c. a) Need to include upregulated genes, too.

We thank the reviewer for this comment, we have made the edit as suggested.

REVIEWER add comment: Figure 5c and d legend do not respond to the new ones. Also, no explanation about two Random lanes. Need to explain the difference between them.

b) Statistical validation is also needed (5b and 5d also).

We thank the reviewer for this comment, we have included random controls and appropriate statistical comparisons.

REVIEWER add comment: The reviewer could not find where appropriate statistical comparisons were included.

7. SINE B2 element targeting

Related to the issue 6, it is also valuable to clarify what determines the specificity of SETDB1 targeting to SINE B2 elements. Any targeting motifs?

We thank the reviewer for this question. Similar to the question asked by reviewer 1, we explored the recruitment of SETDB1 to SINE B2 elements. Integrating KAP-1 ChIP-seq data from WT ESCs (De Iaco et al. 2017), we found that the SINE B2 elements that gain CTCF binding in *Setdb1* KO and SETDB1 Catalytic mutants show higher enrichment levels of KAP-1 (Extended Data Fig. 3d). This suggests that indeed, SETDB1 is recruited by KAP-1. While SETDB1-repressed SINE B2 elements are more significantly enriched with binding motifs for factors including Znf189 (p-value = $1e-1286$), the same motif is also enriched in all SINE B2 elements. Therefore, we cannot conclude which specific KRAB-ZNF functions to recruit the KAP-1/SETDB1 complex to specific elements. This is an interesting focus that can be explored in future studies.

REVIEWER add comment: Regarding "SETDB1-repressed SINE B2 elements are more significantly enriched with binding motifs for factors including Znf189", "SETDB1-repressed SINE B2 elements" are SETDB1-targeted ones or H3K9me3-enriched ones or enhanced CTCF-binding ones after SETDB1 depletion? Should be clarified.

8. P9, line 269-271 "Furthermore, *Rasl11a* gained interactions with a locus that was highly accessible and showed a slight increase of H3K27ac and depletion of H3K9me3 signal in *Setdb1* KO (Fig. 5a)." The reviewer could not recognize which region was described in this sentence. The authors should mark the locus stated in this sentence in Fig. 5a.

We thank the reviewer for pointing this out. In the current submission, we have added labels to better indicate the bins, which change in interactions.

REVIEWER add comment: Still, not clear for me that the enhancement of contact frequency between the loci highlighted with yellow line in *Setdb1* KO. Furthermore, it is also not obvious that the slight increase of H3K27ac and depletion of H3K9me3 signal.

REVIEWER add comment: Although it was not reported at the time of submission of this manuscript, the same finding (SINE B2-SETDB1 targeting and CTCF binding regulation which impact on chromatin 3D genome organization) has been already published in *Genes Dev* (Gualdrini et al., *Genes Dev* 2022). Therefore, it should be briefly described at the end of this manuscript (apart from scoop protection).

Reviewer #2 (Remarks to the Author):

The comments from the previous review have been addressed. I am OK with the revision.

Response to reviewers' Comments:

Reviewer #1 (Remarks to the Author):

The reviewer is satisfied with the major responses of the authors, but would like the following minor points to be corrected. Once this correction is made, there will be no further comments.

1) 5. Fig. 5c. a) Need to include upregulated genes, too.

We thank the reviewer for this comment, we have made the edit as suggested. REVIEWER add comment: Figure 5c and d legend do not respond to the new ones. Also, no explanation about two Random lanes. Need to explain the difference between them.

We apologize for this oversight. We have now updated the figure legends. Regarding the two random controls, they are number matched randomly selected loci. Therefore, we conducted the selection twice to match the upregulated and downregulated gene numbers, respectively. We have added the following sentence to explain the reason for the two "random" lanes. The text now reads as follows:

"The randomly selected genes were picked twice independently (n=660 and n=286), corresponding to the numbers of up- and down-regulated genes, respectively."

2) b) Statistical validation is also needed (5b and 5d also).

We thank the reviewer for this comment, we have included random controls and appropriate statistical comparisons.

REVIEWER add comment: The reviewer could not find where appropriate statistical comparisons were included.

We apologize for being unclear on the location of the addition. We have previously included the statistical comparisons in Extended data figure 5b, c and d. These show the statistical comparison relevant to Figure 5b and 5d. We have now also added the corresponding figure information in the Extended data figure legends.

3) 7. SINE B2 element targeting

Related to the issue 6, it is also valuable to clarify what determines the specificity of SETDB1 targeting to SINE B2 elements. Any targeting motifs?

We thank the reviewer for this question. Similar to the question asked by reviewer 1, we explored the recruitment of SETDB1 to SINE B2 elements. Integrating KAP-1 ChIP-seq data from WT ESCs (De Iaco et al. 2017), we found that the SINE B2 elements that gain CTCF binding in Setdb1 KO and SETDB1 Catalytic mutants show higher enrichment levels of KAP-1 (Extended Data Fig. 3d). This suggests that indeed, SETDB1 is recruited by KAP-1. While SETDB1-repressed SINE B2 elements are more significantly enriched with binding motifs for factors including Znf189 (p-value = 1e-

1286), the same motif is also enriched in all SINE B2 elements. Therefore, we cannot conclude which specific KRAB-ZNF functions to recruit the KAP-1/SETDB1 complex to specific elements. This is an interesting focus that can be explored in future studies.

REVIEWER add comment: Regarding “SETDB1-repressed SINE B2 elements are more significantly enriched with binding motifs for factors including Znf189”, “SETDB1-repressed SINE B2 elements” are SETDB1-targeted ones or H3K9me3-enriched ones or enhanced CTCF-binding ones after SETDB1 depletion? Should be clarified.

To answer the reviewer’s question, “SETDB1-repressed SINE B2” referred to those B2 elements that gained CTCF binding in *Setdb1* KO cells. As mentioned previously, given that the Znf 89 motif was significantly enriched in both this subset of B2 elements and also all B2 elements, we cannot conclude whether Znf189 is involved in recruiting KAP-1/SETDB1. This will indeed be an interesting topic for future investigation.

4) 8. P9, line 269-271 “Furthermore, *Rasl11a* gained interactions with a locus that was highly accessible and showed a slight increase of H3K27ac and depletion of H3K9me3 signal in *Setdb1* KO (Fig. 5a).” The reviewer could not recognize which region was described in this sentence. The authors should mark the locus stated in this sentence in Fig. 5a.

We thank the reviewer for pointing this out. In the current submission, we have added labels to better indicate the bins, which change in interactions. REVIEWER add comment: Still, not clear for me that the enhancement of contact frequency between the loci highlighted with yellow line in *Setdb1* KO. Furthermore, it is also not obvious that the slight increase of H3K27ac and depletion of H3K9me3 signal.

We understand that the reviewer’s critique of the example not being obvious. As such, we have selected another example with much clearer change in H3K27ac and H3K9me3. The new example locus does not change the previous conclusions and we hope serves as a more obvious illustration and representation of the phenomenon. We have added Extended data figures 5h, i, and the following text:

“For example, the *Ccny* gene, which harboured no H3K9me3 at its promoter in WT cells, was upregulated upon SETDB1 depletion. No changes in H3K27ac, chromatin accessibility (Extended Data Fig. 5h), and DNA methylation levels (Data not shown) were detected. Furthermore, the *Ccny* promoter gained interactions with a distal bin, likely through the enhanced CTCF and cohesin binding. This distal region was highly accessible and showed substantial increase of H3K27ac and depletion of H3K9me3 enrichment levels in *Setdb1* KO cells (Extended Data Fig. 5i).”

5) REVIEWER add comment: Although it was not reported at the time of submission of this manuscript, the same finding (SINE B2-SETDB1 targeting and CTCF binding regulation which impact on chromatin 3D genome organization) has been already

published in *Genes Dev* (Gualdrini et al., *Genes Dev* 2022). Therefore, it should be briefly described at the end of this manuscript (apart from scoop protection).

We thank the reviewer for pointing this out. We have now cited this study and added a paragraph to the discussion to highlight the consistencies and differences with our work. The text now read as follows:

“While this manuscript was under preparation, Gualdrini *et al.* reported similar findings, demonstrating that H3K9me3 restricts CTCF recruitment at SINE B2 elements in human macrophages⁷³. With SETDB1 depletion, formation of new chromatin interactions was associated with transcriptional upregulation of specific genes. These results align with our observations, which offer additional insights into the underlying molecular mechanism. This includes the alteration in cohesin binding, the redistribution of nucleosomes, and how this phenomenon functions by the catalytic activity of SETDB1 and is independent of DNA methylation. Taken together, in both human and mouse, SETDB1 appears to play cell-type specific roles in regulating CTCF binding.”

Reviewer #2 (Remarks to the Author):

The comments from the previous review have been addressed. I am OK with the revision.

We thank the reviewer for his/her time and efforts in reviewing our submission.

REVIEWERS' COMMENTS

Reviewer #1 (Remarks to the Author):

The authors have addressed all additional comments from reviewers. There are no further comments.